

# Brant: Foundation Model for Intracranial Neural Signal

**Daoze Zhang**[*]
Zhejiang University
zhangdz@zju.edu.cn

**Zhizhang Yuan**[*]
Zhejiang University
zhizhangyuan@zju.edu.cn

**Yang Yang**[†]
Zhejiang University
yangya@zju.edu.cn

**Junru Chen**
Zhejiang University
jrchen_cali@zju.edu.cn

**Jingjing Wang**
Zhejiang University
wjjxjj@zju.edu.cn

**Yafeng Li**
Nuozhu Technology Co., Ltd.
yafeng.li@neurox.cn

## Abstract

We propose a foundation model named Brant for modeling intracranial recordings, which learns powerful representations of intracranial neural signals by pre-training, providing a large-scale, off-the-shelf model for medicine. Brant is the largest model in the field of brain signals and is pre-trained on a large corpus of intracranial data collected by us. The design of Brant is to capture long-term temporal dependency and spatial correlation from neural signals, combining the information in both time and frequency domains. As a foundation model, Brant achieves SOTA performance on various downstream tasks (i.e. neural signal forecasting, frequency-phase forecasting, imputation and seizure detection), showing the generalization ability to a broad range of tasks. The low-resource label analysis and representation visualization further illustrate the effectiveness of our pre-training strategy. In addition, we explore the effect of model size to show that a larger model with a higher capacity can lead to performance improvements on our dataset. The source code and pre-trained weights are available at: `https://zju-brainnet.github.io/Brant.github.io/`.

## 1 Introduction

Brain signals are electrical impulses that are generated by brain neurons and transmit through neural networks. These signals provide important information about brain activity and can usually be monitored in two ways, namely scalp electroencephalography (EEG) and intracranial electroencephalography (iEEG). The former records the electrical brain activity through electrodes placed on the scalp, while the latter implants intracranial electrodes into brain tissue directly to derive neural recordings. Compared with EEG, iEEG manifests significant advantages by providing more abundant, stereotactic and detailed information about brain wave patterns from deeper brain structures, which has been the mainstream method to obtain deep brain information, and is essential in the therapies for some brain diseases (e.g., Parkinson's disease [1], epileptic seizure [2]).

Modeling intracranial recordings has drawn much research attention, but several issues still remain unresolved. Currently, studies for modeling intracranial recordings are mainly divided into two research lines, namely handcrafted feature based methods [3–8] and deep learning based methods [9–12]. Handcrafted feature engineering requires lots of domain knowledge and may only work on specific tasks. And most deep learning based methods are fully supervised, which relies heavily on

---

[*] Equal contribution.
[†] Corresponding author.

37th Conference on Neural Information Processing Systems (NeurIPS 2023).

labeled data. However, labeling data at scale in medical experiments is often infeasible or expensive, which underscores the importance of maximizing the label efficiency. To overcome these limitations, the paradigm of self-supervised pre-training followed by fine-tuning with few samples can greatly reduce the reliance on labels and enable the model to generalize to various downstream tasks.

Moreover, modeling intracranial recordings requires careful consideration of several key factors. (1) Long-term dependency. Since intracranial neural signals are time series and gradual changes in brain activity may only be captured by the long-period analysis [13], long-term temporal dependency is crucial in modeling intracranial data. (2) Spatial correlation. The electrodes implanted in the brain contain many contacts (also called channels), which are distributed across various brain regions. Due to the fact that brain waves propagate through different brain regions [14], signals recorded from different channels can be spatially correlated, reflecting the underlying neural activity. (3) Time and frequency domains. For neural recordings, the time domain provides information about the amplitude and duration while the frequency domain can reveal underlying oscillatory patterns and rhythms [15]. Therefore, modeling neural signals in both domains can provide information more consistent with the neurophysiological mechanisms [16]. To the best of our knowledge, no existing work on intracranial recordings considers all the three key factors simultaneously.

In view of the unresolved issues above, we propose a foundation model for intracranial neural signal named **Bra**in **N**eural **T**ransformer (Brant). The design of our model takes all the three key factors (i.e., long-term dependency, spatial correlation, time and frequency domains) for intracranial signal modeling into account. Moreover, Brant contains more than 500M parameters and is pre-trained on a large intracranial dataset with 1.01 TB data, which can be adapted to accomplish various downstream tasks. Compared to other existing methods for modeling brain signals, Brant can achieve better performance with far fewer labeled samples, showing the great benefit of our work in medical scenarios. As an off-the-shelf model along with the code and weights, Brant can participate in other medical research and experiments, which alleviates the issue of sample and label efficiency.

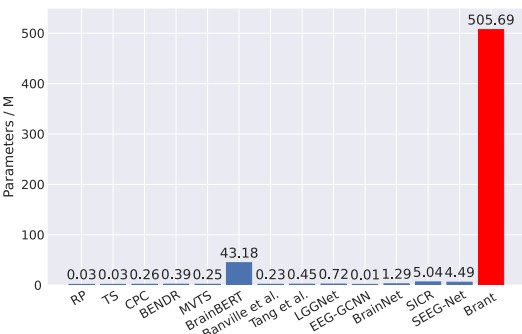

Figure 1: The model scale of existing brain signal models, including RP [17], TS [17], CPC [17], BENDR [18], MVTS [19], BrainBERT [20], LGGNet [21], EEG-GCNN [22], BrainNet [23], SICR [24], SEEG-Net [9], and two works from Banville et al. [25] and Tang et al. [26].

To sum up, the main contributions of our work comprise:

- We propose a foundation model for intracranial neural signals named Brant, which is the largest model on brain signals (shown in Fig. 1) and pre-trained on a large intracranial dataset collected by us, providing a large-scale and off-the-shelf model for medicine.

- To our knowledge, Brant is the first to date that attends long-term dependency and captures spatial correlation across channels, while combining the information from both time and frequency domains.

- Extensive experiments show that Brant generalizes well to various downstream tasks, showing the great potential in neural recordings modeling. Further analysis illustrates the effectiveness of large-scale pre-trained model, demonstrating the medical value of our work.

## 2  Method

**Model overview.** As previously mentioned, we propose Brant to capture long-term dependency and spatial correlation from intracranial recordings, while combining the information from both time and frequency domains. Our model mainly consists of two Transformer encoders, namely, temporal encoder and spatial encoder (shown in Fig. 2). The temporal encoder encodes a sequence of $L$ consecutive patches which focus on the temporal dependency, and the spatial encoder encodes $C$ patches with the same time indices from all channels to capture their underlying spatial correlation.

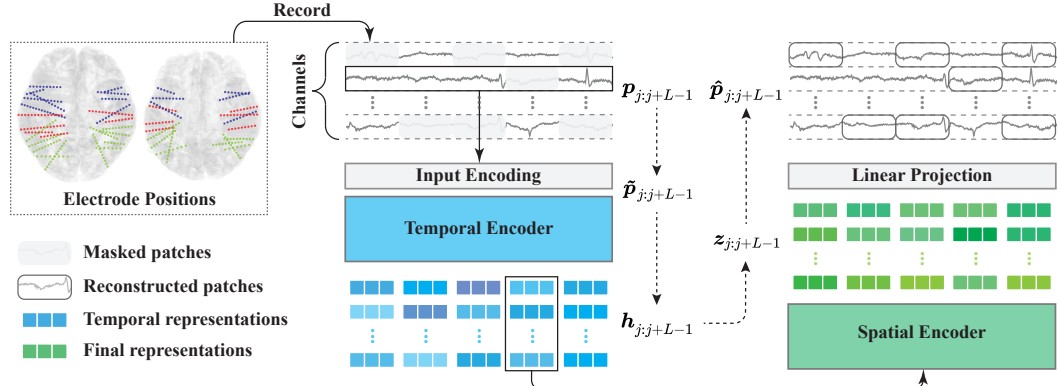

Figure 2: The pre-training framework of Brant. The collected intracranial neural recordings are first processed to a set of patches $\boldsymbol{p} \in \mathbb{R}^{N_p \times C \times M}$ as the input. Then each time we mask a subset of patches $\boldsymbol{p}_{j:j+L-1}$ and map the input patches to the hidden space while adding positional and frequency information to derive the input encoding $\tilde{\boldsymbol{p}}_{j:j+L-1}$. The temporal encoder encodes the input encoding in each channel to derive the temporal representations $\boldsymbol{h}_{j:j+L-1}$. The spatial encoder encodes the temporal representations with the same time indices from all channels to obtain the final representations $\boldsymbol{z}_{j:j+L-1}$. Then a linear head is used to obtain the reconstructed patches $\hat{\boldsymbol{p}}_{j:j+L-1}$ from the final representations.

During pre-training, we randomly mask the input signals recorded from the implanted electrodes, then linearly map the original signals to the latent space and add positional and frequency information through input encoding. Then the input encoding is encoded to latent representations and a linear projection head is added on the top of the spatial encoder which maps the representations to reconstruct the original signals. The details of our model and pre-training task are described as below.

**Patching.** Since neural recordings are electrical signals with high sampling rates, we aggregate timestamps into patches to (1) enhance the locality and extract semantic information; (2) reduce computation and memory usage; and (3) attend a longer temporal dependency [27]. Specifically, given a neural signal $\boldsymbol{x} \in \mathbb{R}^{N \times C}$, where $N$ is the number of timestamps and $C$ is the number of electrode channels, we divide $\boldsymbol{x}$ with length $M$ and stride $S$ to generate a set of patches $\boldsymbol{p} \in \mathbb{R}^{N_p \times C \times M}$, where $N_p = \lfloor \frac{N-M}{S} \rfloor$ is the number of patches in each channel.

**Frequency encoding.** We propose frequency encoding to explicitly inject the information on frequency domain to the observed data. The frequency encoding is mainly based on power spectral density (PSD) which describes the distribution of a signal's total average power over frequency (details about PSD are in App. A) . To be specific, we split the frequency domain into several bands according to the standard description for rhythmic activity [28]: (1) $\theta$ (4-8Hz), (2) $\alpha$ (8-13Hz), (3) $\beta$ (13-30Hz), (4) $\gamma 1$ (30-50Hz), (5) $\gamma 2$ (50-70Hz), (6) $\gamma 3$ (70-90Hz), (7) $\gamma 4$ (90-110Hz), (8) $\gamma 5$ (110-128Hz). For the $i$-th frequency band, a learnable encoding $\boldsymbol{f}_i$ is set as its representation which is shared across all the patches. Then we compute the absolute spectral power of each patch $p_{j,c} \in \boldsymbol{p}, j = 1, ..., N_p; c = 1, ..., C$ in the $i$-th frequency band:

$$P_{j,c}(i) = \log \sum_{\omega \in \text{band}(i)} PSD_{p_{j,c}}(\omega), \quad i \in \{1, 2, \ldots, 8\}, \tag{1}$$

which acts as the weight of $\boldsymbol{f}_i$. The frequency encoding $\mathbf{F}_{j,c} \in \mathbb{R}^D$ of patch $p_{j,c}$ is obtained as the weighted sum of the learnable encodings $\boldsymbol{f}_i$:

$$\mathbf{F}_{j,c} = \sum_{i=1}^{8} \frac{\exp(P_{j,c}(i))}{\sum_{i'=1}^{8} \exp(P_{j,c}(i'))} \boldsymbol{f}_i. \tag{2}$$

**Encoding process.** It contains several steps to encode the input signal to latent representations. Specifically, the input $\boldsymbol{p}_{j:j+L-1} \in \mathbb{R}^{L \times C \times M}$ contains $L \times C$ patches and $\boldsymbol{p}_{j:j+L-1,c} \in \mathbb{R}^{L \times M}$

denotes the patches in $c$-th channel. We first map each sequence of patches $\boldsymbol{p}_{j:j+L-1,c}, c = 1, ..., C$ to the latent space of dimension $D$ by a linear projection $\mathbf{W}_{\text{proj}} \in \mathbb{R}^{D \times M}$, and the projected input will be added with a learnable positional encoding $\mathbf{W}_{\text{pos}} \in \mathbb{R}^{L \times D}$ which monitors the temporal order of patches and the frequency encoding:

$$\tilde{\boldsymbol{p}}_{j:j+L-1,c} = (\mathbf{W}_{\text{proj}} \boldsymbol{p}_{j:j+L-1,c}^{\text{T}})^{\text{T}} + \mathbf{W}_{\text{pos}} + \mathbf{F}_{j:j+L-1,c}, \tag{3}$$

where $\tilde{\boldsymbol{p}}_{j:j+L-1,c} \in \mathbb{R}^{L \times D}$ denotes the input encoding of the original signals $\boldsymbol{p}_{j:j+L-1,c}$. The input encoding will be fed into the temporal encoder to obtain temporal hidden representations $\boldsymbol{h}_{j:j+L-1,c} \in \mathbb{R}^{L \times D}$ and we denote the temporal representations of the whole input as $\boldsymbol{h}_{j:j+L-1} \in \mathbb{R}^{L \times C \times D}$. The spatial encoder further captures the spatial correlation across channels, which takes each of the temporal representations $\boldsymbol{h}_k \in \mathbb{R}^{C \times D}, k = j, ..., j + L - 1$ as the input and outputs the final representations $\boldsymbol{z}_k \in \mathbb{R}^{C \times D}, k = j, ..., j + L - 1$. Overall, given the input signal $\boldsymbol{p}_{j:j+L-1}$, the encoding process returns the corresponding latent representations $\boldsymbol{z}_{j:j+L-1} \in \mathbb{R}^{L \times C \times D}$.

**Self-supervised pre-training.** Self-supervised representation learning is a powerful approach to extract high level abstract representation from unlabelled data. Among those methods to learn representation via self-supervised pre-training, masked autoencoder (MAE) has been proved to be a simple but effective way in many fields [27, 29, 30]. We apply this technique to our self-supervised pre-training, in which the model is trained to reconstruct the whole input given its partial observation.

Given the input patches $\boldsymbol{p}_{j:j+L-1}$, we mask a subset of patches uniformly at random and encode the masked patches to the latent representations $\boldsymbol{z}_{j:j+L-1}$. During the pre-training stage, the representations will be fed into a flatten layer with linear head $\mathbf{W}_{\text{rec}} \in \mathbb{R}^{M \times D}$ to reconstruct the original patches. Finally we utilize an MSE loss to measure the discrepancy between the reconstructed patches $\hat{\boldsymbol{p}}_{j:j+L-1}$ and the original patches $\boldsymbol{p}_{j:j+L-1}$.

## 3 Experimental Setup

### 3.1 Dataset

**Pre-training dataset.** Brant is pre-trained on 1.01 TB neural data, a large clinical intracranial neural signal dataset recorded by stereo-electroencephalography (SEEG) technique from a first-class hospital. The subjects undergo a surgical procedure to implant 4 to 11 invasive electrodes, each with 52 to 153 channels, in their brain. The dataset contains 2528 hours of 1000Hz recordings with more than 1 trillion timestamps. More details are in App. E. We down sample the original signals to 250Hz and generate a set of patches of 6s (1500 timestamps).

**Downstream dataset.** Another neural dataset collected by us with seizure labels is used to fine-tune and evaluate our model. It contains 29.39 GB data with 43 hours of 1000Hz intracranial recordings and we do the same preprocessing (i.e. down sampling and patching) to it as the pre-training dataset. Professional neurosurgeons participate in the labeling of epileptic seizures and the labels consist of two categories, namely, seizure and normal samples. More details are in App. E. For each downstream task, we sample a small subset from the downstream dataset for fine-tuning and evaluation.

### 3.2 Pre-training

For the model configurations, the temporal encoder contains a 12-layer Transformer encoder with model dimension 2048, inner dimension (FFN) 3072 and 16 attention heads, and the spatial encoder contains a 5-layer Transformer encoder with model dimension 2048, inner dimension 3072 and 16 attention heads. During the pre-training, 40% patches in each input sample are masked with zero values uniformly at random. We take 16 input samples as a minibatch and each minibatch contains an average of 24k patches. The model is pre-trained on a Linux system with 2 CPUs (AMD EPYC 9654 96-Core Processor) and 4 GPUs (NVIDIA Tesla A100 80G) for about 2.8 days.

We optimize with Adam [31], updating the model parameters every 4 steps, and the model trains for 750k updates in total. A cyclic scheduler that adopts a basic triangular cycle without amplitude scaling is utilized to adjust learning rate during pre-training. Specifically, we set the basic learning rate as $3 \times 10^{-6}$ and the maximum learning rate as $1 \times 10^{-5}$, then the learning rate steps up (down)

for every 8k updates. We apply mixed precision training with FP32 and BF16 to reduce the memory usage in GPUs for acceleration.

### 3.3 Downstream tasks

As Brant is a foundation model for intracranial recordings, we conduct extensive experiments on several downstream tasks, including short- and long-term signal forecasting, frequency-phase forecasting, imputation and seizure detection, to verify the high capacity of our model in modeling intracranial recordings. We conduct each downstream task for Brant on two settings: (1) fine-tune the model with a learning rate of $1 \times 10^{-7}$; (2) freeze the pre-trained weights. Five random runs were performed to obtain the mean and standard deviation. The detailed setups of these downstream tasks are as follows:

**Short- and long-term signal forecasting.** Predictive observation of the neural signal values is beneficial for the development of warning systems for patients in need of precautionary measures [32, 33]. Therefore, we adopt short- and long-term signal forecasting, in which the learned representations are fine-tuned to predict future signals with different lengths given a past sequence. The past sequence length is set as 15 patches (90s) and the prediction lengths are set as 2 patches (12s) and 20 patches (120s) in short- and long-term forecasting, respectively. We sample 400 minutes of recordings from the downstream dataset, then randomly split into 320 minutes for fine-tuning and 80 minutes for evaluation. A linear prediction head is used to predict the future signals. We adopt MAE and MSE as the performance metrics.

**Frequency-phase forecasting.** In medicine, it is essential to predict the physical features like frequency and phase of brain signals to provide guidance for some treatments. For example, in a therapy that modulates brain activity called *transcranial alternating current stimulation* (tACS), the stimulation control heavily depends on these knowledge about the target brain activity [34]. Therefore, we set up the frequency-phase forecasting task in order for these therapies like tACS to be most effectively used in the treatment of brain disorders [35].

Given a past sequence, this task is to predict the *dominant frequency* and *phase* information (see details in App. B) of intracranial signals in the future. The past and prediction lengths are set as 15 patches (90s) and 5 patches (30s), respectively. We use the same sampling and split strategy as that in the short- and long-term signal forecasting to generate the data for the frequency-phase forecasting task. A linear layer is adopted to predict the dominant frequency and phase of the future signals. As for the metrics, following the work by Mansouri et al. [34], we use MAE for dominant frequency forecasting, and phase locking value (PLV) for phase forecasting. The PLV is a value between 0 and 1 calculated by the equation:

$$PLV = \left\| \frac{1}{T} \sum_{t=1}^{T} \exp(i(\psi(t) - \hat{\psi}(t))) \right\|, \tag{4}$$

where $\psi$ and $\hat{\psi}$ are the original signals and the forecasted signals, respectively.

**Imputation.** During brain signal recordings, measurement problems such as artifact contamination or electrode impairment are not easily corrected, thus the neural recordings will be incomplete. Imputation can fill in these contaminated signals so that other medical devices can keep operating under missing values, making use of available data [36]. For the imputation task, we randomly mask the timestamps in each patch with the ratio of 40% and fine-tune the model to predict the missing values. We adopt the same sampling and split strategy to obtain the imputation data as in the short- and long-term signal forecasting. We add a linear head to make predictions, then apply MAE and MSE as the evaluation metrics to measure the discrepancy between the masked and predicted values.

**Seizure detection.** As one of the most important applications of intracranial recordings, seizure detection task is to evaluate the model ability to distinguish between epileptic seizures and normal waveforms. We sample 250 minutes of recordings from the downstream dataset with about 10% positive (seizure) samples, where 200 minutes are randomly selected for fine-tuning and the remaining 50 minutes for evaluation. An MLP is adopted to classify the pre-trained representations. The evaluation metrics we use are accuracy, precision, recall, $F_1$ and $F_2$ scores. The F-measure is a metric defined as the weighted harmonic mean of precision and recall, with the following equation $F_\beta = \frac{(1+\beta^2) \times precision \times recall}{\beta^2 \times precision + recall}$.

## 3.4 Baselines

As Brant is a pre-training work on neural signals, we extensively compare our model with the advanced self-supervised or unsupervised pre-training works on all the downstream tasks, including works designed for brain signals: RP [17], TS [17], CPC [17], BENDR [18], MVTS [19] and BrainBERT [20]; and for general time series: CoST [37], TF-C [38], PatchTST [27] and TS-TCC [39]. Considering that seizure detection is an important application scenario of intracranial recordings, we further compare our model with several supervised methods which conduct seizure detection on EEG or iEEG to evaluate the performance of Brant in detecting epilepsy. These baselines contain handcrafted feature based methods, including spectral power [28], rhythmicity spectrogram [3] and amplitude-integrated EEG [40]; and the SOTA deep learning based method on seizure detection, including SEEG-Net [9]. More details of the baselines are shown in App. C.

# 4 Experimental Results

## 4.1 Main Results

Fig. 3 summarizes the results of all the downstream tasks. As a foundation model for intracranial recordings, Brant achieves consistent SOTA performance on a variety of tasks compared with other baseline models. We discuss more detailed comparisons of each task in the following paragraphs, where in all the tables we mark values ranking the first (**v**), second (v) and third (*v) in each column.

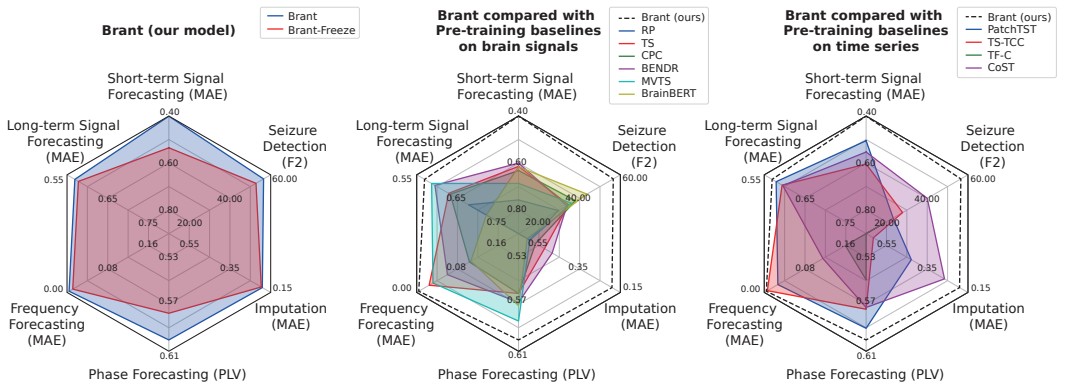

Figure 3: Performance comparison of our model and other baseline models on all downstream tasks. Model performances are plotted on three radar subfigures for clarity with the same coordinate range.

Table 1: Performance on forecasting tasks.

| Model \ Task | Short-term signal forecasting | | Long-term signal forecasting | | Frequency-phase forecasting | |
|---|---|---|---|---|---|---|
| | MAE | MSE | MAE | MSE | ph. PLV | freq. MAE |
| RP [17] | $0.7569_{\pm0.0016}$ | $1.1769_{\pm0.0103}$ | $0.6774_{\pm0.0017}$ | $1.0579_{\pm0.0010}$ | $0.4929_{\pm0.0002}$ | $0.4197_{\pm0.0042}$ |
| TS [17] | $0.6142_{\pm0.0012}$ | $0.8771_{\pm0.0076}$ | $0.6286_{\pm0.0014}$ | $0.9771_{\pm0.0011}$ | $0.5609_{\pm0.0002}$ | $0.0245_{\pm0.0008}$ |
| CPC [17] | $0.6318_{\pm0.0012}$ | $0.8886_{\pm0.0088}$ | $0.6324_{\pm0.0013}$ | $0.9604_{\pm0.0010}$ | $0.5608_{\pm0.0003}$ | $0.1036_{\pm0.0014}$ |
| BENDR [18] | $0.6002_{\pm0.0014}$ | $0.8720_{\pm0.0081}$ | $0.5948_{\pm0.0020}$ | $0.8535_{\pm0.0013}$ | $0.5685_{\pm0.0002}$ | $0.0604_{\pm0.0017}$ |
| MVTS [19] | $0.6868_{\pm0.0019}$ | $1.1859_{\pm0.0102}$ | $0.5867_{\pm0.0026}$ | $0.8381_{\pm0.0014}$ | *$0.5843_{\pm0.0004}$ | $0.0320_{\pm0.0014}$ |
| BrainBERT [20] | $0.6161_{\pm0.0010}$ | $0.8857_{\pm0.0094}$ | $0.7214_{\pm0.0014}$ | $1.4727_{\pm0.0012}$ | $0.5712_{\pm0.0002}$ | $0.1065_{\pm0.0015}$ |
| PatchTST [27] | $0.5050_{\pm0.0010}$ | $0.6482_{\pm0.0062}$ | *$0.5792_{\pm0.0014}$ | *$0.8235_{\pm0.0006}$ | $0.5905_{\pm0.0002}$ | $0.0267_{\pm0.0006}$ |
| TS-TCC [39] | $0.6042_{\pm0.0012}$ | $1.0244_{\pm0.0096}$ | $0.5938_{\pm0.0018}$ | $0.8506_{\pm0.0008}$ | $0.5743_{\pm0.0002}$ | $0.0061_{\pm0.0003}$ |
| TF-C [38] | $1.2530_{\pm0.0025}$ | $2.9441_{\pm0.0146}$ | $1.0683_{\pm0.0030}$ | $2.2491_{\pm0.0016}$ | $0.5493_{\pm0.0004}$ | $0.1593_{\pm0.0029}$ |
| CoST [37] | $0.5531_{\pm0.0011}$ | $1.0922_{\pm0.0092}$ | $0.5955_{\pm0.0012}$ | $0.8546_{\pm0.0005}$ | $0.5728_{\pm0.0003}$ | $0.1152_{\pm0.0018}$ |
| Brant-Freeze | *$0.5352_{\pm0.0008}$ | *$0.7161_{\pm0.0111}$ | $0.5780_{\pm0.0004}$ | $0.7869_{\pm0.0007}$ | $0.5777_{\pm0.0002}$ | *$0.0113_{\pm0.0003}$ |
| Brant | **$0.4007_{\pm0.0014}$** | **$0.4626_{\pm0.0048}$** | **$0.5693_{\pm0.0012}$** | **$0.7676_{\pm0.0005}$** | **$0.6004_{\pm0.0001}$** | **$0.0041_{\pm0.0002}$** |

**Short- and long-term signal forecasting.** The results of short- and long-term signal forecasting are shown on the left and middle of Tab. 1, respectively. Not only does Brant achieve SOTA performance,

but the results of Brant-Freeze are also better than most baselines. Especially on long-term forecasting, Brant-Freeze defeats all the baseline methods, showing the ability to capture long-term dependency of our model. Among the baselines, PatchTST [27] performs well compared to other methods, mainly because it also adopts a patching strategy to attend longer temporal dependency.

**Frequency-phase forecasting.** Forecasting physical features like frequency and phase of neural signal is essential in some medical techniques (see Sec. 3.3). The results of frequency-phase forecasting are shown on the right of Tab. 1, which shows that our model outperforms all the methods, demonstrating that Brant facilitates effective implementation of therapies for some brain disorders.

**Imputation.** As a mitigation for measurement problems, imputation can fill in the incomplete neural signals, allowing other medical devices to continue operating even when data is missing (see Sec. 3.3). From Tab. 2, Brant and Brant-Freeze achieve the best and the second best results among all the methods, which demonstrates the ability of our model in capturing underlying temporal patterns with partially observed neural recordings to imputate the contaminated neural signals.

**Seizure detection.** As seizure detection is an important medical application of intracranial recordings, we not only compare our model with the pre-training based baselines, but also further select 4 supervised method designed for seizure detection. From Tab. 3, Brant outperforms all the other methods and Brant-Freeze achieves second best on most of the metrics. BrainBERT [20] achieves the best accuracy, precision and F2 score among all the baseline methods, primarily due to it provides contextualized neural embeddings and combines the information from time and frequency domains like our model. However, Brant still improves the F2 score by 29.59% over BrainBERT, because our temporal encoder obtains a wider receptive field and the spatial encoder captures the spatial correlation across channels, which are both critical in modeling intracranial recordings.

Table 2: Performance on the imputation task.

| Task Model | Imputation | |
|---|---|---|
| | MAE | MSE |
| RP [17] | $0.5953_{\pm 0.0008}$ | $0.9158_{\pm 0.0038}$ |
| TS [17] | $0.5225_{\pm 0.0008}$ | $0.7476_{\pm 0.0032}$ |
| CPC [17] | $0.5663_{\pm 0.0006}$ | $0.8271_{\pm 0.0034}$ |
| BENDR [18] | $0.4849_{\pm 0.0008}$ | $0.6492_{\pm 0.0031}$ |
| MVTS [19] | $0.5946_{\pm 0.0009}$ | $0.8903_{\pm 0.0032}$ |
| BrainBERT [20] | $0.6083_{\pm 0.0008}$ | $0.9790_{\pm 0.0026}$ |
| PatchTST [27] | $0.4282_{\pm 0.0007}$ | $0.5506_{\pm 0.0012}$ |
| TS-TCC [39] | $0.6144_{\pm 0.0008}$ | $1.0558_{\pm 0.0029}$ |
| TF-C [38] | $1.1876_{\pm 0.0014}$ | $3.0179_{\pm 0.0042}$ |
| CoST [37] | $*0.2652_{\pm 0.0005}$ | $*0.1638_{\pm 0.0007}$ |
| Brant-Freeze | $\underline{0.1963_{\pm 0.0005}}$ | $\underline{0.0865_{\pm 0.0004}}$ |
| Brant | $\mathbf{0.1912_{\pm 0.0003}}$ | $\mathbf{0.0814_{\pm 0.0002}}$ |

Table 3: Performance on the seizure detection task.

| Task Model | Seizure detection | | | | |
|---|---|---|---|---|---|
| | Accuracy | Precision | Recall | F1 | F2 |
| Spectral Power [28] | $89.01_{\pm 0.12}$ | $72.77_{\pm 1.98}$ | $36.07_{\pm 1.23}$ | $48.23_{\pm 0.57}$ | $40.12_{\pm 1.02}$ |
| Rhythmicity Spectrogram [3] | $88.58_{\pm 0.16}$ | $70.31_{\pm 2.09}$ | $37.07_{\pm 1.39}$ | $48.55_{\pm 0.60}$ | $39.10_{\pm 1.09}$ |
| Amplitude-integrated EEG [40] | $88.82_{\pm 0.19}$ | $71.21_{\pm 2.01}$ | $35.73_{\pm 1.50}$ | $47.58_{\pm 0.61}$ | $39.68_{\pm 1.12}$ |
| SEEG-Net | $88.97_{\pm 0.14}$ | $70.45_{\pm 2.20}$ | $38.44_{\pm 1.47}$ | $*49.74_{\pm 0.60}$ | $42.28_{\pm 1.10}$ |
| RP [17] | $67.65_{\pm 1.21}$ | $18.62_{\pm 2.45}$ | $34.71_{\pm 2.11}$ | $24.24_{\pm 1.87}$ | $29.59_{\pm 1.97}$ |
| TS [17] | $85.90_{\pm 0.36}$ | $54.68_{\pm 3.65}$ | $31.66_{\pm 1.97}$ | $40.10_{\pm 0.81}$ | $34.57_{\pm 1.66}$ |
| CPC [17] | $84.72_{\pm 0.40}$ | $48.31_{\pm 2.80}$ | $36.03_{\pm 0.92}$ | $41.28_{\pm 0.66}$ | $37.96_{\pm 1.42}$ |
| BENDR [18] | $88.14_{\pm 0.68}$ | $71.49_{\pm 3.42}$ | $29.83_{\pm 2.03}$ | $42.10_{\pm 1.84}$ | $33.77_{\pm 1.81}$ |
| MVTS [19] | $88.35_{\pm 0.22}$ | $69.43_{\pm 3.19}$ | $32.03_{\pm 2.08}$ | $43.84_{\pm 1.96}$ | $35.90_{\pm 1.94}$ |
| BrainBERT [20] | $*89.59_{\pm 0.12}$ | $77.86_{\pm 3.10}$ | $39.28_{\pm 0.88}$ | $52.21_{\pm 0.39}$ | $*43.60_{\pm 0.98}$ |
| PatchTST [27] | $81.13_{\pm 0.22}$ | $38.71_{\pm 3.64}$ | $21.16_{\pm 2.21}$ | $27.37_{\pm 0.42}$ | $23.27_{\pm 1.26}$ |
| TS-TCC [39] | $88.13_{\pm 0.29}$ | $89.61_{\pm 2.08}$ | $23.81_{\pm 1.50}$ | $37.62_{\pm 0.40}$ | $27.91_{\pm 1.19}$ |
| TF-C [38] | $75.05_{\pm 0.61}$ | $18.75_{\pm 2.41}$ | $19.09_{\pm 2.02}$ | $18.92_{\pm 0.41}$ | $19.02_{\pm 1.24}$ |
| CoST [37] | $81.05_{\pm 0.15}$ | $30.98_{\pm 2.59}$ | $*43.19_{\pm 1.88}$ | $36.08_{\pm 0.79}$ | $40.03_{\pm 1.88}$ |
| Brant-Freeze | $\underline{90.53_{\pm 0.33}}$ | $*77.26_{\pm 3.16}$ | $\underline{48.92_{\pm 0.72}}$ | $\underline{59.87_{\pm 0.62}}$ | $\underline{52.87_{\pm 0.47}}$ |
| Brant | $\mathbf{91.17_{\pm 0.15}}$ | $\mathbf{79.25_{\pm 2.32}}$ | $\mathbf{52.74_{\pm 1.43}}$ | $\mathbf{63.29_{\pm 0.37}}$ | $\mathbf{56.50_{\pm 1.08}}$ |

## 4.2 Model Analysis.

**Low-resource labeled data evaluation.**  In medical scenarios, collecting labeled data for even small experiments is a huge investment. To demonstrate the practical value of our work, we evaluate Brant on seizure detection where the amount of labeled data is limited. Specifically, the pre-trained model is fine-tuned on 200 minutes, 60 minutes and 20 minutes of labeled data that sampled from the downstream dataset (Sec. 3.1), respectively. After fine-tuning, models are evaluated on the same 50 minutes of labeled data which is also sampled from the downstream dataset but non-overlapped with the fine-tuning data.

From Tab. 4, the performances of supervised methods decrease rapidly compared to the self-supervised or unsupervised methods, showing that the representations learned on unlabeled data can improve low-resource settings. Among the pre-training works, our model maintains the most stable performance on 20-minute labeled data. Note that the F2 score of our model on 20-minute labeled data (51.03%) is even higher than the F2 score of the best baseline on 200-minute labeled data (43.60%), demonstrating that Brant fully captures the patterns and semantic information from the intracranial data during pre-training and adapts to downstream tasks more easily.

Table 4: Low-resource labeled data evaluation on the seizure detection task and the relative decrease of the F2 score on 60-minute and 20-minute labeled data versus 200-minute labeled data.

| Model | 200 minutes | 60 minutes | | 20 minutes | |
|---|---|---|---|---|---|
| | F2 | F2 | Decrease | F2 | Decrease |
| Spectral Power [28] | $40.12_{\pm 1.02}$ | $25.95_{\pm 1.92}$ | 35.32% | $5.06_{\pm 1.67}$ | 87.39% |
| Rhythmicity Spectrogram [3] | $39.10_{\pm 1.09}$ | $21.87_{\pm 2.33}$ | 44.07% | $2.81_{\pm 0.87}$ | 92.81% |
| Amplitude-integrated EEG [40] | $39.68_{\pm 1.12}$ | $18.42_{\pm 2.34}$ | 53.58% | $2.05_{\pm 0.95}$ | 94.84% |
| SEEG-Net [9] | $*42.28_{\pm 1.10}$ | $35.54_{\pm 1.90}$ | 15.94% | $12.76_{\pm 2.13}$ | 69.82% |
| RP [17] | $29.59_{\pm 1.97}$ | $27.62_{\pm 2.03}$ | *6.66% | $25.05_{\pm 1.98}$ | 15.34% |
| TS [17] | $34.57_{\pm 1.66}$ | $30.15_{\pm 3.05}$ | 12.79% | $29.61_{\pm 3.34}$ | *14.35% |
| CPC [17] | $37.96_{\pm 1.42}$ | $30.55_{\pm 3.01}$ | 19.52% | $29.57_{\pm 3.74}$ | 22.10% |
| BENDR [18] | $33.77_{\pm 1.81}$ | $25.37_{\pm 3.12}$ | 24.87% | $22.18_{\pm 4.09}$ | 34.32% |
| MVTS [19] | $35.90_{\pm 1.94}$ | $26.62_{\pm 3.11}$ | 25.85% | $24.39_{\pm 4.01}$ | 32.06% |
| BrainBERT [20] | $\underline{43.60}_{\pm 0.98}$ | $\underline{41.93}_{\pm 2.09}$ | $\underline{3.84}$% | $\underline{36.35}_{\pm 3.23}$ | 16.63% |
| PatchTST [27] | $23.27_{\pm 1.26}$ | $18.02_{\pm 2.23}$ | 22.55% | $17.07_{\pm 2.11}$ | 26.64% |
| TS-TCC [39] | $27.91_{\pm 1.19}$ | $25.35_{\pm 2.07}$ | 9.17% | $20.36_{\pm 1.90}$ | 27.05% |
| TF-C [38] | $19.02_{\pm 1.24}$ | $15.97_{\pm 1.23}$ | 16.04% | $13.66_{\pm 2.10}$ | 28.18% |
| CoST [37] | $40.03_{\pm 1.88}$ | $*39.18_{\pm 3.02}$ | **2.12**% | $36.10_{\pm 4.12}$ | $\underline{9.82}$% |
| Brant | $\mathbf{56.50}_{\pm 1.08}$ | $\mathbf{52.30}_{\pm 2.04}$ | 7.43% | $\mathbf{51.03}_{\pm 2.74}$ | **9.68**% |

**Representation analysis.**  As classification task can verify the model capacity in high-level representation learning [41], we visualize the pre-trained representations of Brant and 3 best pre-training based methods on seizure detection task using t-SNE (shown in Fig. 4). Compared to other methods, the representations of seizure and normal signals learned from Brant are separated more clearly during pre-training, which intuitively illustrates our SOTA performance on low-resource label settings. Furthermore, the results explain the good performance of Brant-Freeze on seizure detection.

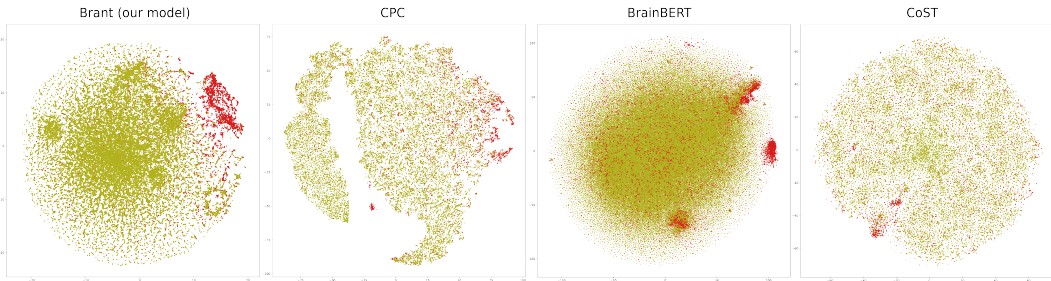

Figure 4:  Pre-trained representation visualization of Brant and 3 best baselines on seizure detection task. The representations of seizure and normal samples are plotted in red (●) and yellow (●).

**Model scale analysis.** To explore the effect of model size on performance, we additionally pretrain three variants of Brant with smaller size: Brant-tiny, Brant-small and Brant-medium (detailed configurations are in App. F) on the same dataset as Brant. All models are pre-trained with an Adam optimizer and a cyclic scheduler. Brant-medium adopts the same learning rate as Brant (Sec. 3.2); for Brant-tiny and Brant-small, the basic and maximum learning rates are $6 \times 10^{-6}$ and $3 \times 10^{-5}$.

We evaluate these smaller size variants on all the downstream tasks (results shown in Fig. 5). As the model size increases, the performances on the downstream tasks show an overall upward trend, where Brant is ahead of the other three variants in all metrics. Also, the decrease in the standard deviation indicates more stable performance for larger models. On such a huge neural dataset, a larger model with a higher capacity results in better generalization ability to a wide range of downstream tasks.

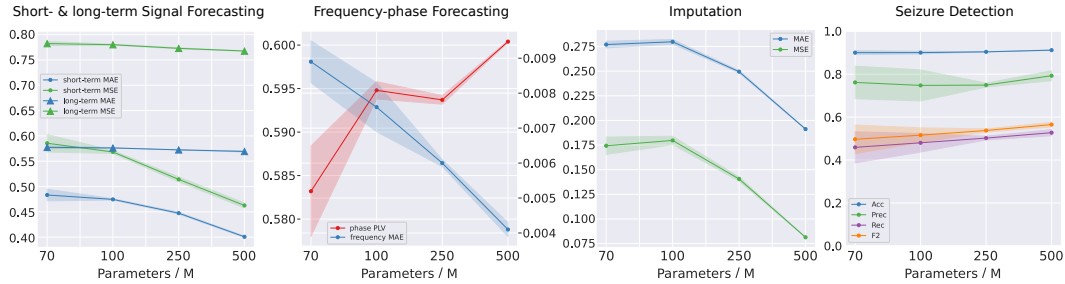

Figure 5: Performance on all the downstream tasks across Brants with different model size: Brant-tiny (70M), Brant-small (100M), Brant-medium (250M) and Brant (500M).

**Ablation study.** We perform ablation experiments to assess the effectiveness of each individual component in our model. Specifically, we remove the following components from our model to examine their respective effects on performance: the temporal encoder (Brant-temporal), the spatial encoder (Brant-spatial), and the frequency encoding (Brant-frequency).

The experimental results of ablation study are given in Tab. 5 and Tab. 6. The results indicate that Brant outperforms other model variants across all metrics, providing evidence for the contribution of each component in our model. Among the different variants, Brant-temporal exhibits the most substantial decrease in performance compared to the full Brant model, emphasizing the significance of long-term dependency in modeling brain signals. The performance degradation observed in Brant-spatial demonstrates the role of considering channel spatial correlation in learning representations of intracranial recordings. Furthermore, the decline of Brant-frequency suggests that simultaneously extracting information from both the time and frequency domains proves beneficial for effectively modeling brain signals.

Table 5: Results of ablation experiments on forecasting tasks.

| Model \ Task | Short-term signal forecasting | | Long-term signal forecasting | | Frequency-phase forecasting | |
| --- | --- | --- | --- | --- | --- | --- |
| | MAE | MSE | MAE | MSE | ph. PLV | freq. MAE |
| Brant-temporal | 0.5356 | 0.7021 | 0.5777 | 0.7876 | 0.5800 | 0.0094 |
| Brant-spatial | 0.5403 | 0.7521 | 0.5827 | 0.7990 | 0.5780 | 0.0092 |
| Brant-frequency | 0.4762 | 0.5676 | 0.5726 | 0.7741 | 0.5645 | 0.0143 |
| Brant | **0.4007** | **0.4626** | **0.5693** | **0.7676** | **0.6004** | **0.0041** |

Table 6: Results of ablation experiments on the imputation and seizure detection tasks.

| Model \ Task | Imputation | | Seizure detection | | | | |
| --- | --- | --- | --- | --- | --- | --- | --- |
| | MAE | MSE | Acc. | Prec. | Rec. | F1 | F2 |
| Brant-temporal | 0.2798 | 0.1797 | 88.56 | 67.39 | 40.25 | 50.40 | 43.78 |
| Brant-spatial | 0.2770 | 0.1741 | 89.93 | **84.77** | 36.88 | 51.40 | 41.57 |
| Brant-frequency | 0.2495 | 0.1406 | 90.16 | 78.41 | 44.03 | 56.39 | 48.26 |
| Brant | **0.1912** | **0.0814** | **91.17** | 79.25 | **52.74** | **63.29** | **56.50** |

**Generalization ability analysis.** To further verify the generalization ability of Brant on more subjects with more heterogeneity, we evaluated the model on data of 31 unseen subjects from two public datasets named MAYO and FNUSA [42]. More details about this study are shown in App. G.

# 5   Related Work

**Intracranial recordings modeling.** With the development of iEEG technique to obtain deep brain information from both cortical and subcortical structures, modeling intracranial recordings has attracted the attention of many researchers. Wang et al. [9] extract time domain features of iEEG signals from multiple receptive fields by utilizing multiscale CNN and LSTM. Guo et al. [10] introduce a hypergraph learning approach to model iEEG signals by detecting high frequency oscillations. Yu and Hu [11] propose EDANN to learn domain-invariant representations of iEEG data from multiple subjects by domain adversarial training. Jiang et al. [8] develop a novel method using short-time resting-state connectivity to identify the seizure onset zone (SOZ) from interictal EEG signals. Wang et al. [12] conduct cross-subject iEEG seizure detection based on adaptive feature fusion of brain network features and single-channel features. However, these works are all supervised that relies heavily on labeled data, which is often difficult and expensive to obtain at scale. Furthermore, most works for intracranial data modeling process each channel independently, ignoring the spatial correlation. Although Chen et al. [23] propose BrainNet which contains a graph diffusion component which measures the brain wave diffusion among channels, their work only focus on the information in time domain and is limited to an individual subject.

**Pre-training on brain signals.** Pre-training on time series [37, 38, 27, 39] has shown good performance in many scenarios (e.g., weather, traffic flow, exchange rates), including some works designed for brain signals. Banville et al. [17] learn representations from EEG signals by a self-supervised temporal context prediction task, revealing clear latent structures related to physiological and clinical phenomena. Kostas et al. [18] address the problem of limited labeled data on EEG by using a contrastive self-supervised learning task to pre-train a model named BENDR, which is then fine-tuned for downstream tasks. Potter et al. [19] propose an unsupervised approach to model EEG signals using a transformer-based model with a signal reconstruction task. Cai et al. [43] propose to study the self-supervised learning framework for brain signals that can be applied to pre-train either scalp or intracranial EEG data. Wang et al. [20] propose a pre-training work named BrainBERT for intracranial recordings and conduct experiments on iEEG data. They adopt time-frequency representations but ignore the spatial correlation between channels, which is critial in modeling intracranial recordings (additional commentary on differences between our work and BrainBERT is in App. D).

# 6   Conclusion

We propose a task-agnostic foundation model, Brant, which learns powerful representations of intracranial recordings. Brant is the largest pre-training model on brain signals, whose design (1) attends a long temporal dependency; (2) captures the spatial correlation between channels; and (3) extracts information from both time and frequency domains. Experimentally, Brant achieves consistent SOTA performance on various downstream tasks w.r.t. medical scenarios. Further analysis shows the effectiveness and benefit of a large-scale pre-trained model in the field of medicine. Brant is an off-the-shelf model with its code and weights, which significantly alleviates the issue of sample and label efficiency and can directly participate in other medical research and treatment.

**Limitations and future works.** By pre-training on a large amount of intracranial data, Brant contains over 500M parameters, far more than other existing works on brain signals. However, compared to other fields such as CV and NLP in which the models can reach billions of parameters and achieve good performance on a variety of tasks by zero-shot learning, there is still potential for further improvement of our work. In the future, by scaling up our dataset, the scale of our model can be further expanded to capture higher-level semantic information from neural data, revealing more complicated brain activities and dynamics, to provide assistance for more healthcare applications.

## Acknowledgments

This work is supported by NSFC (No.62176233), the National Key Research and Development Project of China (No.2018AAA0101900) and the Fundamental Research Funds for the Central Universities.

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

## Ethics Statement

The data collection and experiments conducted in our work have been approved by the Institutional Review Board (IRB) and passed ethical review. All participants have signed informed consent forms.

## A    Intuition of Using PSD as Weights

PSD displays the spectal power distribution in different frequencies of brain signals. The power distribution in different frequency bands is associated with different brain functional states. For example, during wakefulness, $\alpha$ and $\beta$ waves are more active; during sleep, $\delta$ and $\theta$ waves are more prominent; and $\gamma$ waves will become significant during epileptic seizures. Therefore, in medical research, the power distribution in different frequency bands is often used as a feature reflecting the functional state of the brain.

In our work, we treat the power distribution as the weights of learnable representations $f_i, i = 1, ..., 8$ in 8 standard frequency bands. The intuition is that the frequency bands with higher power are more representative of the brain's physiological state, and should be given more weight in their representations. In this way, the model can focus more on the frequency bands where the main power of the brain signals resides.

## B    Details of the Dominant Frequency and Phase

**Dominant frequency.**   Dominant frequency is defined as the frequency related with the greatest power or amplitude in the brain signal. Dominant frequency can be calculated from Fourier Transform spectrum. Mathematically, given a brain signal series $s(t), t \in (t_0, t_1)$, the amplitude $A(\omega)$ on frequency $\omega$ is:

$$A(\omega) = |F(\omega)| = \Big| \int_{t_0}^{t_1} s(t)e^{-i\omega t}dt \Big|, \tag{5}$$

and the dominant frequency $\omega_D$ is:

$$\omega_D = \arg\max_{\omega} A(\omega) \tag{6}$$

**Phase.**   The instantaneous phase of a complex-valued function signal $s(t)$ is the real-valued function:

$$\psi(t) = angle(s(t)), \tag{7}$$

where $angle(\cdot)$ is the angle between the positive real axis and the direction of complex number $s(t)$.

## C    Details of Baselines

Firstly, we compare our model to the existing self-supervised or unsupervised pre-training works on brain signals. The Details of these baseline models are given here:

- RP, TS, CPC: Three self-supervised learning approaches with three self-supervised learning pretext tasks of relative positioning (RP), temporal shuffling (TS) and contrastive predictive coding (CPC) to learn representations of EEG signals based on StagerNet.
- BENDR: A self-supervised training model that learn compressed representations to model completely novel raw EEG sequences recorded with differing hardware, and different subjects performing different tasks.
- MVTS: An unsupervised transformer-based anomaly detection approach to train an autoencoder with a masking strategy on non-seizure signals.
- BrainBERT: A reusable transformer for intracranial field potential recordings enables classifying complex concepts and decoding neural data.

Secondly, we compare our model with the advanced self-supervised or unsupervised pre-training works designed for general time series, including:

- CoST: A time series representation learning framework for long sequence time series forecasting, which applies contrastive learning methods to learn disentangled seasonal-trend representations.
- TF-C: A decomposable pre-training model, where the self-supervised signal is provided by the distance between time and frequency components, each individually trained by contrastive estimation.
- PatchTST: An efficient design of Transformer-based models for multivariate time series forecasting and self-supervised representation learning, based on the segmentation of time series into subseries-level patches.
- TS-TCC: An unsupervised time series representation learning framework to learn robust temporal and discriminative representations by designing a tough cross-view prediction task and a contextual contrasting module.

Furthermore, as seizure detection is the most important application scenario of intracranial recordings, to evaluate Brant's ability in seizure detection, we compare our model with some seizure detection methods on EEG or iEEG. These baselines include handcrafted feature based methods and a deep learning based method:

- Spectral Power: The spectral power in a particular frequency band means the power of a signal in that frequency band. The rhythmic activity on brain signals is typically described in terms of the spectral power or its ratio on some standard frequency bands.
- Rhythmicity Spectrogram: This feature displays a density spectral array of frequency and power characteristics of brain signals, providing a graphical depiction of the amplitude of primary rhythmic EEG components present in four frequency bands.
- Amplitude-integrated EEG: A trace of electrical activity displayed on a semilogarithmic graph of peak-to-peak amplitude over time, which can be used to monitor and diagnose seizure activity.
- SEEG-Net: A model that can address the problems of sample imbalance, cross-subject domain shift, and poor interpretability and realizes high-sensitivity SEEG pathological activity detection. Since the source code of SEEG-Net is not released, we re-implement it to conduct experiments.

## D  Detailed Differences between Brant and BrainBERT

BrainBERT is also a pre-training work conducted on intracranial recordings, but our work is different with them in the following aspects.

**Data and model scale.**  Pre-training often requires a large amount of data by which the model fully learns the distributions, and the pre-trained model usually contains a huge quantity of parameters to fit the data. According to the paper of BrainBERT and the model weights released by its authors [20], the total amount of data and parameters in BrainBERT is 43.7 hours and 43.18M, respectively. Brant is pre-trained on 2528 hours of data and the scale of Brant reaches to 505.69M parameters.

**Method.**  As discussed in the main paper, Brant considers long-term dependency, channel correlation, time and frequency domains when modeling intracranial recordings. BrainBERT provides contextualized neural embeddings and adopts time-frequency representations, but ignores the channel correlation which reflects the propagation of brain waves.

**Downstream tasks.**  BrainBERT is evaluated on four classification tasks related with the audio listened from the subjects, which mainly focus on a particular stimulus, i.e., passive movie viewing. Brant is evaluated on various downstream tasks, including signal forecasting, frequency-phase forecasting, imputation and seizure detection, all of which are medically valuable (described in Sec. 3.3).

## E  Details of the Datasets

**Pre-training dataset.**  The pre-training dataset contains 1.01TB data with a sampling rate of 1000Hz. The details of the pre-training dataset are given in Tab. 7.

**Downstream dataset.** The downstream dataset contains 29.39 GB data with a sampling rate of 1000Hz. The details of the downstream dataset with seizure labels are given in Tab. 8. For each downstream task, we sample a small subset from the downstream dataset for fine-tuning and evaluation. The sampling and split strategies are described in the setup of downstream tasks in Sec. 3.3.

Table 7: Details of the pre-training dataset.

| Subject id | Recording hours | #Electrodes | #Channels | #Sample points |
|---|---|---|---|---|
| 1 | 235.34 | 10 | 124 | 105,055,776,000 |
| 2 | 82.39 | 4 | 52 | 15,423,408,000 |
| 3 | 393.09 | 10 | 120 | 169,814,880,000 |
| 4 | 137.45 | 10 | 133 | 65,811,060,000 |
| 5 | 214.22 | 8 | 116 | 89,458,272,000 |
| 6 | 386.70 | 8 | 101 | 140,604,120,000 |
| 7 | 111.55 | 7 | 67 | 26,905,860,000 |
| 8 | 207.42 | 5 | 47 | 35,095,464,000 |
| 9 | 759.80 | 11 | 134 | 366,527,520,000 |

Table 8: Details of the downstream dataset.

| Subject id | Recording hours | #Seizure timestamps | #Normal timestamps |
|---|---|---|---|
| A | 3.46 | 3,298,000 | 383,154,000 |
| B | 4.63 | 539,000 | 216,288,000 |
| C | 5.23 | 2,129,000 | 563,091,000 |
| D | 6.00 | 3,427,000 | 622,977,000 |
| E | 6.00 | 1,311,000 | 544,090,000 |
| F | 6.00 | 496,000 | 361,282,000 |
| G | 5.92 | 7,278,000 | 243,255,000 |
| H | 5.98 | 3,330,000 | 717,831,000 |

## F   Details of Brant with Different Scales

To explore the effect of model scales on performance, we additionally pre-train three variants of Brant with smaller size: Brant-tiny, Brant-small and Brant-medium on the same dataset. Detailed configurations of Brant and its variants are shown in Tab. 9.

Table 9: Configurations of different scales of Brant.

| Model \ Config | Temporal/Spatial Encoder Layer | Model Dimension | Inner Dimension | Parameter Number |
|---|---|---|---|---|
| Brant-tiny | 8/4 | 768 | 2048 | 68.49M |
| Brant-small | 8/4 | 1024 | 2048 | 103.90M |
| Brant-medium | 12/5 | 1280 | 3072 | 249.22M |
| Brant | 12/5 | 2048 | 3072 | 505.68M |

## G   Details of the Generalization Ability Analysis

As a supplement to the experiments on downstream dataset in the main text, to further verify the generalization ability of Brant on more subjects with more heterogeneity, we evaluated the model on data of 31 unseen subjects from two public datasets named MAYO and FNUSA [42]. The MAYO dataset includes intracranial neural signals of 18 subjects collected from Mayo Clinic (Rochester, Minnesota, United States of America). The FNUSA dataset comprises intracranial data of 13 subjects recorded from St. Anne's University Hospital (Brno, Czech Republic).

Table 10: Performance on the short- and long-term forecasting tasks on two public datasets.

| Task & Dataset Model | Short-term signal forecasting | | | | Long-term signal forecasting | | | |
|---|---|---|---|---|---|---|---|---|
| | MAYO | | FNUSA | | MAYO | | FNUSA | |
| | MAE | MSE | MAE | MSE | MAE | MSE | MAE | MSE |
| RP | 1.3881 | 3.0642 | 1.3786 | 3.0317 | 1.4025 | 3.1277 | 1.3839 | 3.0562 |
| TS | 0.9464 | 1.4572 | 0.9505 | 1.4846 | 0.8318 | 1.1138 | 0.8353 | 1.1213 |
| CPC | 0.8009 | 1.0311 | 0.7922 | 1.0306 | 0.7979 | 1.0254 | 0.7898 | 1.0249 |
| BENDR | 0.9673 | 1.4950 | 0.9507 | 1.4585 | 1.0943 | 2.4782 | 1.0921 | 2.4325 |
| MVTS | 1.1027 | 1.7043 | 1.0838 | 1.6627 | 1.2475 | 2.8251 | 1.2450 | 2.7731 |
| BrainBERT | *0.7971 | 1.0110 | 0.7829 | 0.9971 | *0.7943 | *1.0183 | *0.7872 | *1.0202 |
| PatchTST | 0.7969 | *1.0227 | *0.7879 | *1.0237 | 0.7479 | 0.9815 | 0.7498 | 1.0183 |
| TSTCC | 1.4434 | 2.7430 | 1.4532 | 3.0083 | 1.7217 | 2.7587 | 1.7246 | 2.8697 |
| TFC | 2.1544 | 4.0941 | 2.1689 | 4.4900 | 2.5697 | 4.1174 | 2.5741 | 4.2831 |
| CoST | 0.8812 | 1.2783 | 0.8935 | 1.2895 | 0.7996 | 1.0299 | 0.8094 | 1.0582 |
| Brant | **0.6860** | **0.8064** | **0.6915** | **0.8249** | **0.7329** | **0.8856** | **0.7374** | **0.9118** |

Table 11: Performance on the frequency-phase forecasting task and the imputation task on two public datasets.

| Task & Dataset Model | Frequency-phase forecasting | | | | Imputation | | | |
|---|---|---|---|---|---|---|---|---|
| | MAYO | | FNUSA | | MAYO | | FNUSA | |
| | ph. PLV | freq. MAE | ph. PLV | freq. MAE | MAE | MSE | MAE | MSE |
| RP | 0.2194 | 1.0222 | 0.2232 | 1.0225 | 0.7025 | 0.8340 | 0.7079 | 0.8556 |
| TS | 0.5064 | 0.0501 | 0.5076 | 0.0517 | 0.7381 | 0.8935 | 0.7427 | 0.9215 |
| CPC | *0.5601 | 0.1183 | *0.5616 | 0.1119 | 0.7725 | 0.9623 | 0.7737 | 0.9796 |
| BENDR | 0.5459 | 0.0638 | 0.5486 | 0.0639 | 0.8906 | 1.0880 | 0.9001 | 1.1510 |
| MVTS | 0.5389 | 0.0672 | 0.5397 | 0.0674 | 0.8097 | 0.9891 | 0.8187 | 1.0463 |
| BrainBERT | 0.5075 | 0.0355 | 0.5091 | 0.0356 | 0.7908 | 1.0045 | 0.7812 | 1.0039 |
| PatchTST | 0.5627 | *0.0455 | 0.5640 | *0.0470 | 0.4934 | 0.4753 | 0.5073 | 0.4896 |
| TSTCC | 0.1285 | 0.2756 | 0.1288 | 0.2752 | 0.7411 | 0.9201 | 0.7451 | 0.9488 |
| TFC | 0.0714 | 0.5367 | 0.0712 | 0.5366 | 0.7640 | 0.9486 | 0.7681 | 0.9781 |
| CoST | 0.5279 | 0.0482 | 0.5292 | 0.0484 | *0.4964 | *0.4783 | *0.5088 | *0.4914 |
| Brant | **0.5824** | **0.0304** | **0.5814** | **0.0305** | **0.4887** | **0.4622** | **0.5020** | **0.4835** |

Table 12: Performance on the seizure detection task on two public datasets.

| Task & Dataset Model | Seizure detection | | | | | | | | | |
|---|---|---|---|---|---|---|---|---|---|---|
| | MAYO | | | | | FNUSA | | | | |
| | Acc. | Prec. | Rec. | F1 | F2 | Acc. | Prec. | Rec. | F1 | F2 |
| Spec Power | 58.79 | 38.77 | 44.27 | 41.34 | 43.05 | 58.08 | 38.48 | 46.42 | 42.08 | 44.58 |
| Rhythm Spec | 58.06 | 31.23 | *45.06 | 36.89 | 41.39 | 56.27 | 30.53 | 47.64 | 37.21 | 42.84 |
| aEEG | 58.36 | 31.19 | 43.98 | 36.50 | 40.65 | 57.83 | 36.52 | 38.70 | 37.58 | 38.25 |
| SEEG-Net | 57.92 | 38.38 | 46.72 | *42.14 | 44.77 | 58.21 | 39.12 | *49.29 | *43.62 | *46.85 |
| RP | 86.51 | 46.19 | 20.29 | 28.20 | 22.86 | 70.19 | 59.29 | 19.88 | 29.78 | 22.93 |
| TS | 72.55 | 48.72 | 17.72 | 25.98 | 20.30 | 69.00 | 52.88 | 22.64 | 31.70 | 25.56 |
| CPC | *87.29 | 52.90 | 24.01 | 33.03 | 26.96 | 68.34 | 50.40 | 24.98 | 33.40 | 27.78 |
| BENDR | 80.95 | 23.95 | 21.11 | 22.44 | 21.62 | 62.28 | 5.70 | 34.04 | 9.76 | 17.06 |
| MVTS | 69.19 | 53.30 | 24.99 | 34.03 | 27.96 | 67.11 | 49.66 | 19.68 | 28.19 | 22.38 |
| BrainBERT | 85.39 | 44.13 | 44.92 | 44.52 | *44.76 | 68.31 | 44.96 | 49.50 | 47.12 | 48.52 |
| PatchTST | 87.39 | 53.02 | 29.42 | 37.84 | 32.29 | *69.94 | *55.81 | 26.08 | 35.55 | 29.19 |
| TSTCC | 77.26 | 10.47 | 9.16 | 9.77 | 9.40 | 63.83 | 24.40 | 11.36 | 15.50 | 12.71 |
| TFC | 82.09 | 11.41 | 4.91 | 6.87 | 5.54 | 68.71 | 29.36 | 5.05 | 8.62 | 6.05 |
| CoST | 87.39 | *53.02 | 29.42 | 37.84 | 32.29 | 65.92 | 36.34 | 33.65 | 34.95 | 34.16 |
| Brant | **89.40** | **58.78** | **70.80** | **64.23** | **68.02** | **83.51** | **83.60** | **54.18** | **65.75** | **58.28** |

The results of generalization study are given in Tab. 10, Tab. 11 and Tab. 12. Compared to other baseline models, Brant holds the consistent SOTA performance on these downstream tasks on the unseen 31 heterogeneous subject data collected from different hospitals in different countries, highlighting the generalization ability of our model.

