# OpenReview forum: "Brant: Foundation Model for Intracranial Neural Signal"
_NeurIPS.cc/2023/Conference — NeurIPS 2023 poster_

### Official Review · Reviewer_uVSw · 2023-07-02

**Soundness:** 3 good
**Presentation:** 3 good
**Contribution:** 2 fair
**Rating:** 6
**Confidence:** 4

**Summary:**

This paper proposes a foundation model (called BPT) for modeling intracranial brain signals. BPT consists of a Transformer encoder for temporal encoding, and a separate Transformer encoder for spatial encoding. BPT is pre-trained for reconstructing masked input signals, and is fine-tuned for a variety of downstream tasks (short-term and long-term forecasting, seizure detection etc.). Experimental results suggest that BPT outperforms other self-supervised/unsupervised pre-training models, as well as supervised models on downstream tasks. Model scaling analysis further suggests the benefit of having a large model size.

**Strengths:**

1. While the ideas of self-supervised pre-training using a reconstruction task and modeling brain signals using Transformer encoders are not new, the originality of this work comes from a large model pre-trained (largest model for brain signals to date) on a large dataset, as well as its analyses on a variety of downstream tasks.
2. The methods are technically sound, and the experiments are carefully designed to support the authors’ claims.
3. The paper is relatively easy to understand.

**Weaknesses:**

1. Baselines seem to be chosen selectively. Some existing works are shown in Figure 1 but not included as baselines (e.g., GNN-based methods, including BrainNet, EEG-GCNN, Tang et al. [23]).
2. While the pre-training dataset is large (1.1TB), it only consists of 9 subjects (Appendix E), which could limit the representativeness of the learned latent representations.
3. There is an overload of notations in the Method section.
4. The authors claim that BPT is a foundation model for brain signals. However, as I noted above, the pre-training dataset only comprises iEEG data from 9 subjects. The downstream experiments are also done on iEEG data from a small number of subjects (8 subjects). In my opinion, more diverse pre-training data and more downstream experiments on larger datasets are needed to demonstrate that BPT can be served as a foundation model.

**Questions:**

1. Notations: In “Model overview” section, $L$ is used to denote the number of patches. But later in “Patching” section, $N_p$ is used to denote the number of patches. Please be consistent.
2. What’s the intuition of frequency encoding using PSD as weights? Please justify in “Frequency Encoding” section.
3. The patch length P is projected to dimension D (“Encoding Process” section) using a linear layer. Doesn’t this ignore temporal dependency within the patch? In my opinion, this is a limitation and should be addressed (e.g., using a small CNN layer) or discussed.
4. The patch length is selected to be 6s. Please justify this choice.
5. Please include some ablation studies to demonstrate the effectiveness of each component in BPT. The ablation studies can be centered around the 3 claimed advantages: 1) long-range dependency, 2) spatial correlation, and 3) time and frequency domains.
6. Is the data split patient-wise? If not, the downstream tasks may be relatively easy as the same patient’s signals are seen during fine-tuning.
7. For the imputation task, it is unsurprising that BPT would perform well given that it’s pre-trained for a masked reconstruction task. Instead of imputing masked timestamps, a more challenging imputation task would be to mask an entire channel and impute the values in the masked channel.
8. Please also include GNN-based approaches listed in Figure 1 as baselines. e.g., BrainNet, EEG-GCNN, Tang et al. [23].


**Limitations:**

Limitations about the dataset size are discussed. In my opinion, the following two points are also limitations and should be discussed:
- Data come from a small number of subjects.
- Projecting the temporal dimension of a patch using a linear layer.

---

> ### Author Rebuttal · Authors · 2023-08-09
>
> Thank you for your constructive and detailed comments. Responses to specific comments are listed below.
>
> **1. Q1: Clarification on notations**
>
> Thank you for pointing out this potential ambiguity. Actually, $N_p$ denotes the number of all the patches from an entire channel, while $L$ denotes the number of patches input to the temporal encoder each time. We will provide a clearer clarification in lines 92-94 in the manuscript.
>
> **2. Q2: Intuition for using PSD as weights**
>
> PSD displays the spectal power distribution in different frequencies of brain signals. The power distribution in different frequency bands is associated with different brain functional states. For example, during wakefulness, $\alpha$ and $\beta$ waves are more active; during sleep, $\delta$ and $\theta$ waves are more prominent; and $\gamma$ waves will become significant during seizures [R1]. Therefore, in medical research, the power distribution is often used as a feature reflecting the functional state of the brain.
>
> In our work, we treat the power distribution as the weights of 8 learnable representations in 8 standard frequency bands. The intuition is that the frequency bands with higher power are more representative of the brain’s physiological state, and should be given more weight in their representations. In this way, the model can focus more on the frequency bands where the main power of the brain signals resides. We will explain this intuition clearer in line 104 of the manuscript.
>
> **3. Q3&L2: Discussion on whether to extract temporal dependencies within the patch**
>
> Thank you for the insightful suggestion. For high-frequency signals like intracranial data, we aggregate timestamps into patches to enhance the locality and extract semantic information. Here we consider the patch as the smallest unit for representing brain signals, so we ignore the temporal dependency within the patch. Moreover, recent works [27,R2] in the field of time series also project patches to the hidden dimension using a linear layer, and achieves SOTA performance on a variety of time series data. Therefore, we choose to use a linear layer to project patches to dimension D.
>
> Inspired by your valuable suggestion, we agree that using CNN to perform convolution on patches is beneficial in extracting the internal dependencies within patches. We will include this discussion in the the limitation of the manuscript.
>
> **4. Q4: Reason for choosing a patch length of 6s**
>
> Based on recent works [R3,17] and the experience of our neurologists, setting a few seconds as the minimum unit is empirically a reasonable choice to capture the underlying physiological states of the brain from neural signals. For instance, in [R3], researchers used window lengths of 10s and 6s, respectively, to learn the spatiotemporal dynamics of the brain. In [17], segment length of 5s was chosen on intracranial signals for their model pretraining. Therefore, we choose 6s as the patch length for training of our model.
>
> **5. Q5: Supplementary ablation experiments**
>
> We apologize for initially overlooking the detailed justifications for the model architecture design. To demonstrate the effectiveness of each components in BPT, we did the ablation experiments on three model variants you mentioned. The ablation results are given in Table 4 in the PDF in Global Response. We will include the ablation study in the appendix of our manuscript.
>
> **6. Q6: Generalization experiment addresses the concern of split-subject setting**
>
> Thank you for the comment. In the experiments in our manuscript, the data for fine-tuning and testing were sampled from the same group of subjects. We hope that the additional experiments on 31 new subjects in the Global Response can provide complementary support for the generalization of our model on split-subject setting.
>
> **7. Q7: Discussion on masking an entire channel in the imputation task**
>
> Thank you for the suggestion. For measurement problems, imputation can fill in the incomplete neural signals, allowing other medical devices to continue operating even when data is missing [36] (details in lines 188-191). Based on our experience in data acquisition and cleaning processes, missing records are usually partial sample point missing caused by mechanical or electromagnetic interference, rather than an entire channel lost. Moreover, in the field of time series, existing works [41,R4] on the imputation task were also randomly masking a certain proportion of sample points to simulate malfunctions in real-world systems. Depending on the application and following existing works, we impute the masked sample points as our imputation task.
>
> **8. Q8: The reason why GNN-based methods cannot be baselines**
>
> Thank you for the suggestion. These GNN-based methods were not included in the baselines, not because we selectively chose the baselines, but because these graph-based methods require constructing graph nodes using channels (each channel corresponds to a node). However, the number of channels varies among different subjects, which means that graph-based methods can only be applied to individual subjects. Therefore, we apologize for not being able to include these methods as baselines because they cannot fulfill our experimental setting on multiple subjects.
>
> If you remain unconvinced that we have sufficiently justified a higher score, please inform us where we can further improve our work or any concerns you have. Thank you!
>
> Reference:
>
> [R1] P. A. Abhang, et al. Chapter 2: Technological Basics of EEG Recording and Operation of Apparatus, 2016
>
> [R2] Shao, Zezhi, et al. Pre-training enhanced spatial-temporal graph neural network for multivariate time series forecasting. 2022
>
> [R3] Zhang Y, et al. Functional annotation of human cognitive states using deep graph convolution. 2021
>
> [R4] Zhou T, et al. One Fits All: Power General Time Series Analysis by Pretrained LM. 2023

---

> > ### Comment · Reviewer_uVSw · 2023-08-13
> >
> > Thank you for the detailed responses and additional experiments. I have increased my score.

---

> > > ### Author Response · Authors · 2023-08-15
> > > **Thank you**
> > >
> > > We sincerely appreciate your effort in helping us to enhance the paper and your support for our work.

---

### Official Review · Reviewer_5rAo · 2023-07-05

**Soundness:** 4 excellent
**Presentation:** 4 excellent
**Contribution:** 3 good
**Rating:** 7
**Confidence:** 2

**Summary:**

This paper presents Brain Pre-trained Transformer (BPT) for intracranial neural signal modeling. The model was trained on a large-scale dataset and achieved SOTA performance on multiple downstream tasks.

**Strengths:**

- originality: the authors developed a transformer model for brain signal modeling, which can learn rich representations from intracranial neural recordings.
- quality: this paper presents the largest pre-training model on brain signals, which can capture long temporal dependency and spatial correlations.
- clarity: the paper is well-written and organized.
- significance: the model achieved consistent SOTA performance on various downstream tasks.

**Weaknesses:**

I'm not an expert in this field. So I'm not able to give constructive comments on the weakness.

As a foundation model for medical scenarios, it seems that the interpretability analysis is missed.

**Questions:**

Will the datasets be publicly available?


**Limitations:**

It would be great to provide the interpretability analysis of the learned features.

---

> ### Author Rebuttal · Authors · 2023-08-09
>
> Thank you for your careful reading and comments. We are happy that you acknowledged that our model can learn rich representations from intracranial neural recordings. Responses to specific comments are listed below.
>
> **1. Q1: Our planned steps for the dataset release**
>
> Thank you for the valuable suggestion. We apologize for the temporary unavailability of the dataset, because it involves highly sensitive intracranial data and personal privacy concerns.
>
> In the future, in order to make this dataset a scientific research tool and better serve the research community, we plan to promote the release of our dataset in the following three stages:
>
> - Stage1: We plan to publicly release the hidden representations learned by our model through pretraining, along with their corresponding labels, shortly after our work is accepted. This will enable other researchers to not only replicate our model’s experimental results, but also utilize these representations and labels for other research of interest.
>
> - Stage2: We will actively communicate with the hospital and aim to release the raw data of a portion of subjects by the end of the year, to support further research efforts. The same as the work in our manuscript, these releases will be also conducted in compliance with ethical review requirements.
>
> - Stage3: In the future, we will explore the possibility of releasing the full dataset following approval of the relevant ethical review, to allow researchers to use the large-scale dataset for more research.
>
> **2. Ethics statement**
>
> The data collection and experiments conducted in our work have been approved by the Institutional Review Board (IRB) and passed ethical review. All participants have signed informed consent forms.
>
> If you still have any other new concerns, we would be eager to know what we can do to address any questions or concerns you may have. Thanks for your thoughtful feedback again!

---

> > ### Comment · Reviewer_5rAo · 2023-08-14
> >
> > Thanks for your explain very much. I keep my previous suggestion: accept.

---

> > > ### Author Response · Authors · 2023-08-15
> > > **Thank you**
> > >
> > > We truly appreciate your effort in helping us to strengthen the paper and your support for our work.

---

### Official Review · Reviewer_T6RH · 2023-07-06

**Soundness:** 3 good
**Presentation:** 3 good
**Contribution:** 4 excellent
**Rating:** 7
**Confidence:** 4

**Summary:**

The authors propose a foundation model named BPT for modeling intracranial neural signals. BPT is pre-trained on a large corpus (1.01 TB) of intracranial recordings collected by the authors. Animportant feature is that BPT is designed to capture long-term temporal dependency and spatial correlation from neural signals, combining information in both time and frequency domains. BPT is pre-trained using a masked autoencoder objective. The results claim that  BPT achieves state-of-the-art performance on various downstream tasks.


**Strengths:**

- The paper proposes a novel self-supervised foundation model for modeling intracranial neural recordings.
- The authors claim that the size of the model is of unparalleled precedent.
- The experiments are thorough, well-designed, and provide strong empirical evidence for the effectiveness of the proposed method.
- The paper is clearly written, with a well-structured flow of ideas.
- The work has high potential impact and significance for the medical field and perhaps a step towards future neural interfaces.
- The model will be made publicly available.

**Weaknesses:**

- The work is scaling up previous architectures. No
- if possible it would be great to see model preformance on more clinically meaningful tasks
- The model analysis is a bit superficial. Would be useful for the reader to have a more in-depth analysis, e.g. how different frequency bands contribute to the model performance, how the learned representations capture spatial or temporal relationships in iEEG signals, etc.
- The results are not compared with other SOTA models on the same datasets. Despite this may not be feasible, specially on medical domain.

**Questions:**

- The pre-training task adopted in this work is masked autoencoder. Why did the authors choose this task over other self-supervised tasks (e.g. contrastive learning, denoising autoencoder)? A comparison and analysis on different pre-training tasks would be helpful.

- Patients demographics: The dataset seems to contain data from many subjects. The analysis of the dataset does not provide much details on the subjects information (e.g. age, gender, diagnoses). More details on this would help to assess generability of the model.

- Is it possible to compare with humans for the same tasks?

**Limitations:**

- Lack of external validation, despite not being feasible due to the lack of dataset.
- The authors should discuss any potential privacy concerns or negative societal impacts as neural data is highly sensitive and building large models from such data raises privacy concerns.

---

> ### Author Rebuttal · Authors · 2023-08-09
>
> Thank you for your constructive and detailed comments. We are excited that you felt our work has high potential impact and significance for the medical field and perhaps a step towards future neural interfaces. Responses to specific comments are listed below.
>
> **1. Q1: Discussion on why not use other SSL tasks (e.g. contrastive learning, denoising autoencoder)**
>
> Thank you for the insightful question. The motivation is mainly driven by the variety of downstream tasks. As noted in a recent review on self-supervised learning [R1], generative self-supervised learning (like autoencoder) has the ability to recover the original data distribution without assumptions for downstream tasks, which enables generative models’ wide applications both classification and generation. In contrast, contrastive learning has assumed the downstream applications to be classifications [R1]. Therefore, as a representative of the generative approach, autoencoders meet our needs to design the architecture of a foundation model, which is expected to generalize to various downstream tasks well.
>
> As for the denoising aotuencoder, both masked autoencoder and denoising autoencoder share the same goal of learning a robust representation of the input signal. Therefore, we empirically believe that there is no intuitive difference between using masked autoencoder or denoising autoencoder in our model architecture. Following exsiting works such as masked language model (MLM) in natural language processing and PatchTST in time series [27], masked autoencoder is used to build our model.
>
> **2. Q2: Notes on patient demographics & Generalization experiment for generability**
>
> Thank you for the thoughtful suggestion. Due to the restrictions from the hospital, we sincerely apologize for the inability to disclose certain subject profile details such as age, gender, and diagnoses due to concerns regarding personal privacy. As a supplement to the individual diversity information of subjects in our dataset, we can only disclose that the age range of subjects spans from infants to adults, including males and females. The dataset comprises a collection of relatively diverse and heterogeneous subjects.
>
> To further alleviate concerns regarding generability of our model, we conducted additional experiments (in Global Response) on 31 new subjects from two public datasets, MAYO and FNUSA. As shown in Table 1 to Table 3 in the PDF in Global Response, BPT holds the consistent SOTA performance on these downstream tasks, demonstrating the generability of our model.
>
> **3. Q3: Discussion on comparing with humans for the same tasks**
>
> Thank you for the valuable suggestion. We agree that comparing models to humans is a good way to evaluate the effectiveness of models in assisting humans. However, it is worth noting that the successful completion of these downstream tasks by humans relies on the expertise of professionally trained and experienced neurologists. Conducting direct comparisons between machines and humans requires recruiting specialized neurologists to perform these tasks, which is challenging for us to carry out currently. We appreciate your valuable suggestion and will consider incorporating human comparisons in our future works. Thank you again.
>
> If you have any other concerns, please let us know what we can do to address any concerns you may have. Thanks for your thoughtful feedback again!
>
> Reference:
>
> [R1] X. Liu et al. Self-Supervised Learning: Generative or Contrastive, in *IEEE Transactions on Knowledge and Data Engineering*, vol. 35, no. 1, pp. 857-876, 1 Jan. 2023.

---

> > ### Comment · Reviewer_T6RH · 2023-08-14
> >
> > Thank you for addressing the general concerns and my questions. I maintain my suggestion of accepting the paper.

---

> > > ### Author Response · Authors · 2023-08-15
> > > **Thank you**
> > >
> > > We sincerely appreciate your effort in helping us to enhance the paper and positive recognition of our work.

---

### Official Review · Reviewer_5spB · 2023-07-06

**Soundness:** 3 good
**Presentation:** 3 good
**Contribution:** 3 good
**Rating:** 6
**Confidence:** 5

**Summary:**

This manuscript trains a foundation model for intracranial signals based upon a very large electrophysiological dataset.  The methods are based upon a transformer architecture with adaptations to account for the electrode structure and important frequency properties in electrophysiological systems.  The manuscript present experiments compared to many baselines on the tasks, include forecasting, imputation, and seizure detection.

Update after rebuttal:
Most of my concerns have been addressed on the extensive rebuttal, and I have therefore adjusted my score.  Some of the weaknesses remain, such as the limited demonstration of downstream tasks, which is limited to seizure detection, and the original method for seizure detection not using split subjects.  In the final version, these limitations should be noted.  Additionally, the method for calculating uncertainty should be set to report inter-subject variability, and the approach used should be included in the revised manuscript.

**Strengths:**

This type of electrophysiological signals are of high importance in many fields, including several types of electrophysiological signals.  This manuscript could contribution to an exciting line of work building foundation models that are capable of enhancing neuroscience, neurology, neurosurgery, and psychiatry.  The need for foundation models in this area is huge, as studies are frequently limited by relatively small sample sizes, usually insufficient to capture complex patterns appearing in the brain.

The ideas are presented in a straightforward way, and the application is clear.

In my view, the primary contribution is simply to run these algorithms at such a large scale, which is critically necessary to start building real foundation models.

I was impressed by the number of comparison algorithms that were run.

**Weaknesses:**

While I am enthused by the ideas put forth in this manuscript, I view several claims in this manuscript as exaggerated or unsupported.  First, it is necessary to note that while the data scale is huge, the number of individuals is not.  The training data is made up of $N_{train}=9$ individuals, and the test set is made up of $N_{test}=8$ individuals.  As the primary source of heterogeneity in many tasks is the differences between individuals, it is important to assess the variance and performance over individuals rather than windows (e.g., see [1]).  The reported uncertainties seem entirely too small to be over 8 individuals, and those metrics need to be recalculated to more accurately reflect the performance and evidence for the method.

In a similar vein, it is typical in seizure detection to separate out individuals for the training and test set, whereas this manuscript uses data from the same individuals, as described in lines 198-200.  This overestimates performance.  Notably, the mentioned methods do cross-subject validation (e.g. your reference 9).

One of the claims is that this method is much improved over competing methods, but the training data for methods was different.  As mentioned in Supplemental Section C, BrainBERT was pretrained on a tiny fraction of the data that BPT was trained on.  As such, it is difficult to assess whether the performance improvement was due to greater amount of training data (as well as better matched training data, as inter-device comparisons can be challenging).  This should be clarified by training methods on all the same data and with similar schemes.  Likewise, the other comparisons were not trained on the same data, making it difficult to assess whether the performance improvement is due to methodological improvement or simply better matched training data (e.g., EEG-> intracranial recordings is non-trivial).

The claims in the manuscript is that this method is shown to generalize well to several medical scenarios, but I do not see evidence to support that.  The only medical scenario that I see is on seizure, whereas the other tasks are not directly medical scenarios.  Additionally, there are several limitations of this claim: (1) trained on only a single hospital, whereas hospital and device differences can significantly shift signals; (2) total number of individuals in the data set is very small; and (3) the comparisons are not especially clear since they were trained on alternative datasets.  I would encourage the authors to precisely evaluate what evidence of generalization they have provided.


[1] Vu, Mai-Anh T., et al. "A shared vision for machine learning in neuroscience." Journal of Neuroscience 38.7 (2018): 1601-1607.

**Questions:**

Is the performance improved due to larger and better matched training data, or because the method is inherently better?

Once you account for the fact that there are only 8 individuals in the test set, how strong is the evidence of improvement?



**Limitations:**

The limitations were not properly addressed; please see weaknesses.

---

> ### Author Rebuttal · Authors · 2023-08-09
>
> Thank you for your constructive and detailed comments. We apologize for the imprecise claims in certain parts of our manuscript and misunderstandings caused by our writing. We will reconsider the claims to be more precise. We hope that the responses below could address your specific comments.
>
> **1. Q1&W3: Clarification for the fairness of training data and comparison**
>
> Thank you for pointing out this ambiguity. We regret not clearly explaining the following fact in our manuscript: To ensure the fairness of our experiments, all pretraining based baselines are re-pretrained on our pretraining dataset, instead of utilizing the weights published (if any) by the original work. In other words, all the performance comparison are **conducted on the same data** and with the similar schemes, including the BrainBERT and other baselines designed for EEG. Therefore, the performance gains are due to our better model rather than larger and better matched training data.
>
> One thing worth noting is, in Supplemental Section C, we describe that "The total amount of data in BrainBERT is 43.7 hours". What we mean here is that the model weights released by the authors of BrainBERT are pretrained on data of 43.7 hours.
>
> We will clarify these facts more clearly in line 214 and 509 of our revised manuscript.
>
> **2. Q2&W1: Generalization experiment addresses the concern of subject number**
>
> Thank you for the valuable comment. Our response includes two parts:
>
> (1) Obtaining intracranial recordings is challenging as it requires craniotomy surgery for electrode implantation, which involves extensive protocols and approvals. Therefore, in the field of intracranial neural signals, the scale of the subject number in datasets has not reached that of EEG datasets. For instance, a recent work published on ICLR'23, BrainBERT, also included only 10 subjects, with an average recording duration of 4.37 hours per subject [17].
>
> Although the work in the manuscript includes only 9 pretraining subjects, the large data scale is still helpful for the model’s generalization ability. This is because intracranial data is a non-stationary time series that encompasses numerous variations, which is sufficient to cover the relatively most common types of epilepsy seen in clinical practice.
>
> (2) We agree that, with a test set of 8 subjects, the experimental results are not that strong to support the improvements of our model. To illustrate the generalization ability of our model on more subjects with more heterogeneity, we added new experiments on the data of another 31 subjects in Global Response. We hope that it can provide an evidence for the improvement of our model on performance involving individual heterogeneity.
>
> **3. W1: Clarification on the calculation of uncertainty**
>
> Thank you for the suggestion. We indicate in lines 162-163 that the reported standard deviations were obtained from five random runs, with each evaluation randomly sampling data from 8 subjects. So the uncertainties are not calculated from the experimental results over 8 individuals.
>
> **4. W2: Generalization experiment addresses the concern of split-subject setting**
>
> Thank you for the constructive comment. Our response includes two parts:
>
> (1) We concur that the experimental settings that separate training and test subjects (split-subject setting) are highly appropriate in epilepsy detection. Actually, there also exists some seizure detection works without this separation, such as [20,23,R1]. We apologize for the setting we chose, which has limited our experiments to fully showcase the effectiveness of our model on subjects not involved in training.
>
> (2) Following the comment, we added new experiments on 31 subjects in Global Response. We hope that this experiment will help alleviate your concerns regarding performance overestimation and model generalization.
>
> **5. W4: Clarification on the claims about medical scenarios**
>
> Thank you for the valuable comments. As an emerging field, the application of intracranial data in machine learning for healthcare is currently focused on seizure detection. In addition to seizure detection, which is directly relevant to medical scenarios, intracranial data may also involve some indirectly related situations, such as tACS, warning systems development, missing data recovery, etc. Therefore, by adding relevant downstream tasks, we hope to provide potential assistance and contribute to the research in these areas.
>
> Among these downstream tasks, PatchTST performs well on the forecasting, CoST excels on the imputation, and BrainBERT performs best on the seizure detection. However, only BPT achieves consistent SOTA performance on all these downstream tasks, which demonstrates that our model can generalize well to these downstream tasks.
>
> In line 73, we describe that "BPT generalizes well to various downstream tasks w.r.t. several medical scenarios". Actually, what we intended to claim here is that our model can generalize to these downstream tasks, but there might be some imprecision in our wording. We will clarify the relationship between our work and these medical scenarios more clearly in the manuscript. Thank you again.
>
> **6. W4: Generalization experiment addresses the concern of hospital and device diversity**
>
> Thank you for the thoughtful suggestion. We hope that the added experiments in Global Response can convince you that our model can generalize to more subject data collected from different hospitals with different devices.
>
> If you remain unconvinced that we have sufficiently justified a higher score, we would be eager to know where we can make further improvements or address any concerns you have. We appreciate your feedback!
>
> Reference:
>
> [R1] Y. Zhao, et al. Patient-Specific Seizure Prediction via Adder Network and Supervised Contrastive Learning, in *IEEE Transactions on Neural Systems and Rehabilitation Engineering*, vol. 30, pp. 1536-1547, 2022.

---

> > ### Comment · Reviewer_5spB · 2023-08-15
> > **Thank you for your response**
> >
> > Most of my concerns have been addressed, and I have revised my review accordingly.

---

> > > ### Author Response · Authors · 2023-08-16
> > > **Thank you**
> > >
> > > We wholeheartedly appreciate and deeply cherish your efforts in helping us to strengthen the paper and your recognition of our work.

---

### Author Rebuttal · Authors · 2023-08-09

# Global Response to AC and all reviewers
Thanks to all reviewers for the careful reading and thoughtful feedback. We are grateful for acknowledging "the model size is of unparalleled precedent" and our work "has high potential impact and significance for the medical field and perhaps a step towards future neural interfaces" (Reviewer T6RH). We are also excited that our work "could contribute to an exciting line of work building foundation models that are capable of enhancing neuroscience, neurology, neurosurgery, and psychiatry." (Reviewer 5spB). Further, we are delighted to hear "the originality comes from a large model pretrained on a large dataset, as well as its analysis on a variety of downstream tasks" (Reviewer uVSw) with "consistent SOTA performance on various downstream tasks" (Reviewer 5rAo).

In response to reviewers' comments, we performed two new experiments: generalization experiment and ablation experiment, summarizing the additional studies.

## About Generalization Experiment

To supplement the generalization ability of BPT on more subjects with more heterogeneity, we added the new generalization experiment on the data of another 31 previously useen subjects, as a solution to similar concerns raised by some reviewers. These concerns include:

- the small number of test subjects (Reviewer 5spB, uVSw),
- the lack of separation between fine-tuning and testing subjects (Reviewer 5spB, uVSw), and
- the single-source of the hospital and device (Reviewer 5spB).

We introduce the generalization experiment as follows:

- Subject number: We evaluated the model on data of 31 unseen subjects from two public datasets named MAYO and FNUSA [R1]. These two public datasets include 18 and 13 subjects, respectively.

- Diversity on countries, hospitals and devices: These 31 subjects were from two distinct hospitals or institutions located in different countries. As for the acquisition equipment, these data were recorded using completely different acquisition systems and electrode types. Details are given in the table below:

  | Dataset | Country & location | Hospital   | Acquisition system   | Electrode type | Electrode material  | Contact diameter | Contact length | Contact distance |
  | ------- | ---------------------------------------------- | ------------------------------ | ---------------------------------------------- | --------------------------------------- | ---------------------------------- | ---------------- | -------------- | ---------------- |
  | MAYO    | Rochester, Minnesota, United States of America | Mayo Clinic                    | Neuralynx Cheetah system (Neuralynx Inc., USA) | depth electrodes (AD-Tech Corp., USA)   | 4 or 8 contact Pt/Ir electrodes   | 1 mm             | 2.3 mm         | 5 or 10 mm       |
  |         |       |        |                 | grids and strips electrodes             | Pt/Ir discs                        | 4.0 mm           | /              | 10 mm            |
  | FNUSA   | Brno, Czech Republic                           | St. Anne’s University Hospital | BrainScope system (M&I, BrainScope, Czech)     | depth electrodes (Temis Health, France) | 5, 10 and 15 contact Pt electrodes | 0.8 mm           | 2 mm           | 1.5 mm           |

- Split-subject setting: To separate out subjects for the training and test set, we re-evaluated the performance of BPT on the 31 unseen subject data. To ensure the fairness of the experiment, for BPT and each baseline method, we followed the same setting, i.e., fine-tuning on our test set of 8 subjects and re-evaluating them on 31 subject data. In this way, we could assess the model’s generalization ability on data of subjects not involved during the training phase.

- Consistency: To maintain consistency with the experiments in our manuscript, we utilized the BPT weights that were fine-tuned on our test set. In other words, the model used for the generalization experiments was identical to the one in our manuscript.

The results of our additional generalization experiment are given in Table 1 to Table 3 in the new PDF uploaded with this Global Response, where we mark values ranking the first (bold), second (underlined) and third (starred) in each column. As shown in Table 1 to Table 3 in the new PDF, compared to other baseline models, BPT holds the consistent SOTA performance on these downstream tasks on the **unseen 31 heterogeneous subject data** collected from **different hospitals in different countries with different devices**, highlighting the generalization ability of our model **on the split-subject setting**. We hope that this additional experiment will help alleviate the reviewer's concerns regarding performance overestimation and model generalization. We will add this new experiment to our revised paper. Its experimental setup and analysis will be added to the Experiment section of the main paper, and its results will be added to Appendix to maintain the 9-page limit.

## About Specific Responses

We have individually addressed all of your comments below, specifically addressing each reviewer’s concerns in the corresponding responses. Please note that in our responses, references in the format "[R1]" indicate citations that are newly added in the rebuttal, while references in the format "[1]" are citations from the original manuscript.

We have dedicated significant effort to improving our manuscript, and we sincerely hope that our responses will be informative and valuable. We would love to receive your further feedback.

Reference:

[R1] P. Nejedly et al. Multicenter intracranial EEG dataset for classification of graphoelements and artifactual signals, *Sci Data*, vol. 7, no. 1, Art. no. 1, Jun. 2020.

---

### Decision · Program_Chairs · 2023-09-21

**Decision:**

Accept (poster)

**Comment:**

The paper has convinced the 4 reviewers. Although the methodology employed is not particularly novel (pretraining with masking task) the experiments have been considered overall convincing with extensive comparisons. It also worth pointing out that this work uses a unique dataset for the field. One can just regret that such data cannot be shared, although the code and the trained models are made available. I am endorsing this work for publication at NeurIPS 2023.